# Tourism and Development: The Impact of Sustainability—Comparative Case Analysis

**Pablo Juan Cárdenas-García** * and **Alejandro Alcalá-Ordoñez** †

Department of Economics, University of Jaen, 23071 Jaén, Spain
* Correspondence: pcgarcia@ujaen.es; Tel.: +34-9532-122-10
† PhD Student in University of Jaen.

**Abstract:** In recent years, tourism has established itself as one of the most important economic sectors worldwide. Given the economic importance of this activity, different international organizations have decided to bet on tourism as a development tool. However, the expansion of tourism, on many occasions, can cause significant environmental deterioration, so it is necessary to analyze the costs and benefits generated by tourism in those territories that host this activity. To deepen our knowledge about the relationship between tourism, environmental sustainability and economic development, the objective of this work is to analyze the relationships that may exist between these three variables, which will allow us to determine if tourism influences economic development, and if there are factors pertaining to environmental sustainability that influence this relationship. In this paper, a comparative analysis of cases is used to analyze how environmental sustainability influences the relationship between tourism and economic development. The analysis carried out shows that tourism, although with some limitations, can be configured as an instrument of economic development.

**Keywords:** tourism; economic development; environmental sustainability; tourism policy





## 1. Introduction

Currently, the economic literature recognizes tourism as an activity that enables economic growth. In many countries, it is considered as the first or second economic industry in terms of economic impact and generation of employment. In fact, tourism is an activity that has great importance in many countries, to the point of becoming consolidated as a significant pillar of the economic activity of some territories [1].

In this sense, there are many international organizations that have highlighted the importance of tourism for economic growth. This activity generated, in 2019 (the year before the start of the COVID-19 pandemic), 10.5% of world GDP and 10.2% of employment [2]. After the impact of the health crisis caused by COVID-19, which has been a significant challenge for the sector due to restrictions on mobility and the insecurity perceived by tourists, tourism has generated, in 2021, 5.8% of world GDP and 5.4% of employment, so the pre-pandemic level has not been recovered [3].

Therefore, tourism has a significant economic impact on economic growth. Because of drag capacity with other economic sectors, generation of jobs, generation of foreign currency, etc., some national and international institutions have begun to defend the capacity of tourism to improve the living conditions of the resident population [4].

However, in recent years, a new trend has begun to develop that emphasizes the important limitations that tourism faces as a tool for improving the socioeconomic conditions in which people live [4]. Among these limitations, tourism is extremely sensitive to the sustainability of tourism development processes, which in the last two decades has become a central element of the debate about the role of tourism as an instrument for generating wealth, employment and, ultimately, economic development [5–8].

The objective of this paper is to identify, through an analysis of cases, if the factors related to the environmental dimension of sustainability can be considered determining

factors in the transformation of tourism growth in a given country into economic development. In this context, through the analysis of case studies, very interesting conclusions can be obtained.

Analyzing the capacity of tourism as an instrument of economic development is a fundamental issue for the efficiency of public policies. For the investments in tourism that different international organizations have been making to be profitable in terms of increased economic development, it is necessary to be clear about the relationship between tourism and development, and if there are environmental factors that favor or hinder this relationship.

## 2. Theoretical Framework

Economic growth via tourism is considered an instrument to improve economic development, since the development of tourism produces an increase in certain indicators—production of goods and services, investment or tax collection, the progress of the population, such as via average income, education or health—through which an improvement can be achieved [9].

On the other hand, some works have begun to appear that question the role of tourism as an instrument of economic development, due to the important limitations that this relationship presents concerning the environmental dimension of sustainability [10].

### 2.1. Tourism and Economic Development

Tourism is considered an activity capable of promoting the global growth of the economy due to its complementarity with other economic activities, i.e., its contribution to GDP, job creation, foreign exchange generation, etc. [1,10]. However, the true importance of tourism, in addition to contributing to the growth of the economy, is the ability for tourism-induced economic growth to influence the economic and sociocultural progress of society, with an improvement in the quality of life of the population [11,12].

However, it is necessary to highlight the negative effects of maintaining a high dependence on tourism as a development instrument, as it greatly increases the vulnerability and fragility of this improvement process; therefore, policies must be developed that allow maximum use of the beneficial effects that tourism brings in order to improve the socioeconomic conditions of the population [13].

In addition, the relationship between tourism and economic development is not an automatic process; some countries that have opted for tourism as an instrument of economic development have not experienced this relationship because the development of tourism requires certain preconditions for it to be successful [4,10].

Indeed, the transformation of tourism growth into an improvement in economic development requires the existence of certain factors that allow the economic growth brought about by tourism to produce the structural transformation of the economic system of a society [14]. The work developed by the United Nations Development Program establishes a series of measures to use tourism to improve the living conditions of host societies [15]. Among these factors, the following could be cited [16–18]: growth rate, redistribution of wealth, freedom of trade, attractiveness, job creation and efficiency in the management of natural resources.

The works that have endorsed this current link between tourism and economic development support this relationship, since the areas analyzed have certain characteristics that enable the growth of tourism activity that leads to an improvement in the socioeconomic conditions of the population.

However, various studies have pointed out different cases in which tourism does not contribute to improving the living conditions of the resident population, among which we can mention [19,20] the existence of leaks, ownership of capital abroad, use of foreign labor or the environmental degradation and cultural stress caused by tourism.

Therefore, although the expansion of tourism has a series of advantages from economic, environmental and sociocultural points of view that present it is an instrument of progress,

the experience of several countries has shown that, without adequate planning, there is a series of risks that prevent tourism from contributing to socioeconomic improvement and even generate significant disadvantages for the receiving society [21].

Therefore, a series of situations that hinder the contribution of tourism to economic development have also been identified.

### 2.2. Economic Development and Environmetal Sustainability

The concept of sustainable economic development has been consolidated worldwide as an inescapable objective, difficult to question. However, despite the fact that it is a purpose of general acceptance, there is still tremendous confusion in its conception and, above all, in the meaning of its multiple dimensions, which makes it difficult to achieve greater operability.

It was not until 1987, in the document known as the Brundtland Report, that the concept of sustainable development formally emerged: "*Meet the needs of present generations without compromising the possibilities of future generations to meet their own needs*" [22] (p. 46). More recently, the United Nations defined "The 2030 Agenda for Sustainable Development", which contemplates a set of 17 sustainable development goals that, in turn, includes a total of 169 goals, indivisible and balanced with the three dimensions of sustainable development, namely, economic, social and environmental [23].

Although the concept of sustainable development includes three dimensions, the analysis in this work will focus on the environmental dimension of sustainability, given that it is the one that is more conceptually delimited and the one for which more information is available.

In this sense, the European Commission defines the environment as: "*the combination of elements whose complex interrelationships form part of the environment, environment and living conditions of the individual and society, as they are or as they feel*" [24] (p. 15).

The environment—identified in this work in a restrictive way with that of natural capital—is essential for the economic development of a society. Thus, a compatible process between any economic activity and the preservation of biodiversity and ecosystems must be configured as a vital factor for economic growth and the improvement of the conditions of the entire society [8].

In effect, in this same line, The 2030 Agenda for Sustainable Development places special emphasis on the sustainable management of natural resources, establishing the need to take urgent measures to protect of natural resources and against the process of climate change [23].

Therefore, it is evident that the environmental dimension of sustainability has a notable influence on the improvement of the socioeconomic conditions of a country, due to the influence it exerts on some specific aspects of economic development [10].

### 2.3. Tourism and Environmetal Sustainability

Tourism has been configured in the 21st century as one of the fastest growing sectors worldwide, which is why, in addition to its economic contribution, it is recognized as a key element for the protection of the environment [25]. However, the increasing number of people moving from their place of origin to tourist destinations poses important challenges related to sustainable development in the receiving countries.

Therefore, the Sustainable Development Goals (SDGs) include some specific targets related to tourism: "8.9. By 2030, devise and implement policies to promote sustainable tourism that creates", "12.b. Develop and implement tools to monitor sustainable development impacts for sustainable tourism that creates jobs and promotes local culture and products" and "14.7. By 2030, increase the economic benefits to small island developing States and least developed countries from the sustainable use of marine resources, including through sustainable management of fisheries, aquaculture and tourism" [23].

In this sense, and to a greater extent, national governments, while they have opted for tourism as a foundation on which to base their economic growth policy, have begun

to include in their policies different objectives related to consumption patterns and more environmentally sustainable production. Indeed, those responsible for tourism policies are currently fully aware of the need to develop tourism in a sustainable way through a greater commitment to the environmental dimension of sustainability [26].

Therefore, when betting on tourism as an economic activity, it is also necessary to develop policies that take into account the implementation of the environmental components of national tourism policies, which entails the need to take additional measures on the use and the conservation of natural resources by the tourism sector, which will lead to the improved positioning of tourism with respect to the environmental dimension of sustainability [25].

Among these measures, aspects such as the reduction of polluting emissions, the improvement of air, soil and water quality, the conservation of biodiversity and the sustainable uses of the Earth are key developing a kind of tourism that is respectful of the environmental dimension of sustainability [26].

### 3. Materials and Methods

The objective of this paper, which consists of analyzing different empirical works that have been carried out in the scientific literature about tourism, economic development and environmental sustainability, is to demonstrate that tourism, accounting for certain environmental factors, can be considered as an instrument of economic development. For this, a methodological structure is followed, similar to that used by Sousa et al. (2022) in their bibliometric review [27].

#### *3.1. Search Strategy*

Regarding the search for relevant case studies, this search was carried out in June 2022 in the two databases, WoS and Scopus, in which the journals with the greatest scientific recognition are indexed. The search on these two platforms was carried out using the keywords "tourism", "economic development" and "environmental sustainability" included in the title, abstract and keywords. In addition, a search was carried out in these databases for works published between 2000 and 2021.

#### *3.2. Criteria for Selection of Articles*

Given that there was a large number of articles that analyzed the relationship between tourism and economic growth, which is not the objective of this review, as well as of exclusively theoretical articles, exclusion criteria were established, specifically: (i) articles that, despite using the term economic development, were actually analyzing economic growth; and (ii) articles that did not include an empirical analysis of the relationship between tourism and economic development.

#### *3.3. Data Collection Process*

The articles that were selected in a first phase were then screened using the PRISMA method (Figure 1), which consists of a set of items based on obtaining evidence for the development of a systematic review through a list checklist and a flowchart [27]. This process resulted in a total of 6 valid articles, which were used as case studies in this review work.

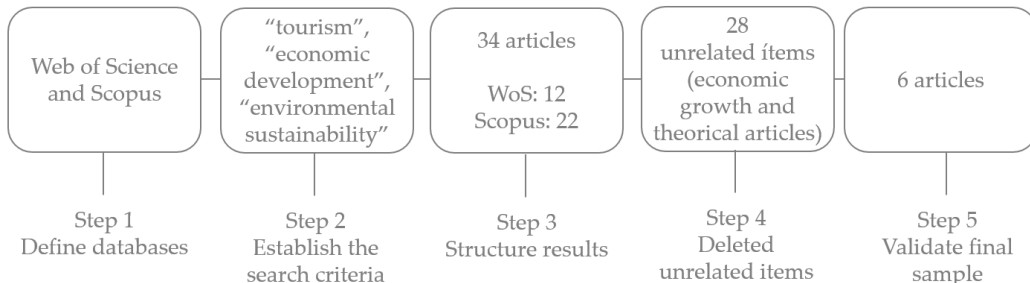

**Figure 1.** Collection and selection process. Source: adapted from Sousa et al. (2022) [27].

### 3.4. Objectives of the Articles Review

This theoretical review, in which common points of all the evaluated works are analyzed, intends to reach the following specific objectives:

- To justify the potential of tourism to enable a real process of economic growth in the countries where this activity takes place, and thus be considered a strategic resource on which to lay the foundations of countries' economic policies;
- To identify the existing characteristics pertaining to environmental sustainability in those receiving areas in which tourism expansion has not given rise to an improvement in the economic development of the population, hence to guide economic policy towards ameliorating the factors that limit this relationship.

## 4. Results

In this section, we proceed to review different empirical works that analyze the relationship between tourism and economic development, taking into account the importance of factors related to environmental sustainability in this relationship.

### 4.1. Tourism Growth Can Influence Economic Development

Below contains a summary of the works that exist in the scientific literature and in which it is shown that tourism growth can influence economic development, Table 1.

**Table 1.** Summary of case analysis (Tourism growth can influence economic development).

| Tourism Growth Can Influence Economic Development | |
| --- | --- |
| Case study | 1 |
| Author | Pulido, Cárdenas, Villanueva (2013) |
| Reference | [28] |
| Destination | 144 countries |
| Period Analyzed | 1991–2010 |
| Statistical Analysis | Econometric modeling |
| Case study | 2 |
| Author | Balsalobre, Leitão (2020) |
| Reference | [29] |
| Destination | EU-28 |
| Period Analyzed | 1995–2014 |
| Statistical Analysis | Panel unit root tests |
| Case study | 3 |
| Author | Sharif et al. (2020) |
| Reference | [30] |
| Destination | China |
| Period Analyzed | 1978–2017 |
| Statistical Analysis | QARDL model |

### 4.1.1. Case Study 1: On the Role of Environmental Sustainability in the Transformation of Tourism Growth into Economic Development

The objective of this work is to identify, through the analysis of some variables, whether the factors related to the environment can be considered as determinants in the transformation of tourism growth into economic development in the countries analyzed. The results obtained in this work show that there are specific factors pertaining to the environmental dimension of sustainability that determine whether tourism growth is transformed into economic development.

Specifically, in the most developed countries, there is a transformation of tourism into an improvement in the living conditions of the population due to the absence of factors that limit this relationship. However, in developing countries, the existence of little land dedicated to cultivation, low use of energy without polluting emissions and excessive use of water resources are factors that are becoming an obstacle for tourism growth to contribute to economic development.

### 4.1.2. Case Study 2: The Role of Tourism, Trade, Renewable Energy Use and Carbon Dioxide Emissions on Economic Growth: Evidence of Tourism-Led Growth Hypothesis in EU-28

The objective of this paper, in addition to other analyses carried out, is to analyze the relationship between the development of tourism, economic growth and the effect of renewable energy, using the countries of the European Union as the geographical scope of analysis. The results achieved in this work show that the increase in the number of tourists received, as well as the development of renewable energies in tourist destinations, favor economic growth. On the other hand, the results of the work also confirm the hypothesis of growth driven by tourism (TLGH) for the data panel that has been used.

The results of this work confirm the TLGH in the cases of the countries analyzed (EU-28) and suggest that public policies should focus on improving tourism infrastructure so that the development of tourism can be used as a policy with which to achieve the sustainable development objectives that have been set by the EU.

### 4.1.3. Case Study 3: Revisiting the Role of Tourism and Globalization in Environmental Degradation in China: Fresh Insights from the Quantile ARDL Approach

The objective of this work is to analyze how the development of tourism, with the arrival of a greater number of tourists, together with economic growth and the associated globalization process, influence—if indeed they do—the process of environmental degradation of tourist destinations. Therefore, the impact of tourism and economic growth on environmental degradation is analyzed, and the results obtained in this work allow the design of sustainable developments within the tourist destination. The results achieved show that economic growth influences environmental degradation, although it is also highlighted that tourism, properly planned, exerts positive environmental externalities.

As for promoting sustainable tourism through the promotion of green energy, industries related to tourism should also enter into this relationship, given that this situation could help activities related to tourism improve the environmental dimension.

### 4.2. *Tourism Growth Does Not Enable Economic Development*

Below contains a summary of the works that exist in the scientific literature and in which it is shown that tourism growth does not enable economic development, Table 2.

**Table 2.** Summary of case analysis (Tourism growth does not enable economic development).

| Tourism Growth Does Not Enable Economic Development | |
|---|---|
| Case study | 1 |
| Author | Aguilera, Bernal, Quintero (2006) |
| Reference | [31] |
| Destination | Colombia |
| Period Analyzed | 1995–2005 |
| Statistical Analysis | Analysis of qualitative data |
| Case study | 2 |
| Author | Raza et al. (2017) |
| Reference | [32] |
| Destination | United States |
| Period Analyzed | 1996–2015 |
| Statistical Analysis | Wavelet transform framework |
| Case study | 3 |
| Author | Pulido, Cárdenas, Espinosa (2019) |
| Reference | [33] |
| Destination | 139 countries |
| Period Analyzed | 2007–2016 |
| Statistical Analysis | Structural equation modelling |

4.2.1. Case Study 1: Tourism and Development in the Colombian Caribbean. Working Documents on Regional Economy

The objective of this work is to demonstrate that, although tourism is one of the most dynamic economic sectors worldwide, in the case of Colombia, the economic growth generated by this activity has not translated into an improvement in the socioeconomic conditions of the population. The authors have identified in this work a series of factors that hinder the contribution of tourism to economic development, including factors related to the environment. Among the negative impacts of tourism, environmental deterioration due to the excessive arrival of tourists and the over-exploitation of certain natural resources and infrastructures stand out.

Therefore, there are factors that negatively affect the relationship between tourism and economic development, among which it is worth highlighting polluted and neglected beaches, environmental carelessness or pollution in the main tourist cities. Therefore, an improvement in the endowment of these factors would favor the improvement of the living conditions of the population residing in tourist areas thanks to the tourist activity generated.

4.2.2. Case Study 2: Tourism Development and Environmental Degradation in the United States: Evidence from Wavelet-Based Analysis

The objective of this work is to analyze the possible relationship that exists between the increase in the number of tourists received by a tourist destination and the environmental pollution caused by the development of tourism. Specifically, this paper analyzes the influence of tourism development on environmental degradation in a tourist destination with a high number of international tourists received. The results of this work show that the increase in the number of tourists received has a unidirectional influence on the process of economic growth (TLGH). The authors also findthe unidirectional causal influence of economic development on environmental degradation, both in the short term as well as in the long term.

Therefore, according to the results of this work, a series of recommendations are generated to guide public policies, with the aim of reducing or limiting the environmental damage that is generated when the number of tourists received increases.

### 4.2.3. Case Study 3: Does Environmental Sustainability Contribute to Tourism Growth? An Analysis at the Country Level

The objective of this work is to analyze whether the environment of the countries analyzed is configured as a relevant variable that influences the likelihood of tourism becoming a tool for economic development. The results of this work determine that a greater importance of tourism for the receiving countries translates into an environmental degradation of the tourist destination; however, some environmental variables influence whether becomes a tool for economic development.

Specifically, there is a double relationship: an increase in tourism degrades the tourist destination from an environmental point of view, but if the destination manages to maintain adequate regulation of environmental sustainability, then tourism can become a tool for economic development. These results are in line with the results that have previously been revealed by some international institutions linked to tourism, as well as with the scientific literature.

## 5. Discussion

According to the case studies analyzed, it is evident that some empirical studies determine that tourism can be used by countries to improve the economic development of their population, although the existence of certain factors is necessary for this relationship between tourism and development to be achieved. In addition, other case studies analyzed in this work determine that an increase in the number of tourists received does not influence the improvement of the living conditions of the population, and that this is due to the existence or absence of factors pertaining to environmental sustainability that influence this relationship.

The common link between both currents: (i) tourism can influence economic development, and (ii) tourism does not allow economic development, is demonstrated in this work as the existence, or lack thereof, of factors pertaining to environmental sustainability that influence the relationship between these two variables. Indeed, according to the studies analyzed, the process of transforming tourism into economic development is not automatic; these factors are required by tourist destinations.

Thus, with the analysis of cases carried out in this work, an approach has been made to the environmental factors that influence the extent to which tourism generated an increase in economic development in tourist destinations. Specifically, on the one hand, the factors that allow tourism to become an improvement in economic development have been identified: reduction in the consumption of energy from natural resources, increase in the use of renewable energies and reduction of polluting emissions. On the other hand, other factors have been identified that make it difficult for tourism to improve economic development in tourist destinations: the overexploitation of agricultural land, the increase in the production of electricity from fossil resources and the increase in the consumption of renewable resources from sweet water.

## 6. Conclusions

It is concluded that there are certain factors that condition the relationship between tourism and economic development. Therefore, there is no single and universal position in the relationship between tourism and economic development; that is, tourism does not allow, in all situations, for countries to improve their level of economic development, nor is it true that it cannot improve development in some specific destinations; this relationship depends precisely on the factors that have been identified in this paper.

Before betting on tourism as an instrument of economic development, and before investing in projects that finance tourism with the aim of improving the living conditions of the resident population, it is necessary that new renewable energies be developed in advance in the destination; these will contribute to the reduction of polluting energies, if tourism is to be used as an instrument of economic development.

These factors that have been identified are in line with the factors that, from a theoretical point of view, were already revealed by the UNWTO when it highlighted as aspects to take into account in the relationship between tourism and development the importance of impact areas such as: reduction of CHG emissions, reduction of air, soil and water pollutants, conservation of biodiversity and sustainable use of land. All of these are factors linked to the SDGs.

In view of these results, the agents involved in tourism development should properly manage the factors related to the environmental dimension of sustainability through policies and actions that allow the preservation of natural resources, given that these factors favor the link between tourism and economic development. In summary, it could be concluded that tourist activity not only exerts an important influence on the economic growth of those areas in which it is develop; it can also be configured as an element to improve the living conditions of the population if the factors related to the environmental dimension of sustainability are properly managed.

Therefore, in the present work, it has been determined that there are factors related to the environmental dimension of sustainability that permit aspects tourism growth to contribute to economic development.

However, there are some limitations in the review work carried out and, therefore, in the conclusions that have been indicated. These limitations stem, primarily, from the small number of empirical works that have analyzed the relationship between tourism, economic development and environmental sustainability, which means that the few existing works have important differences in terms of the time horizon analyzed, the territorial scale or the dimensions of the territories analyzed. Therefore, as a future line of research, it would be interesting to produce new works of empirical content that take advantage of the gap in the scientific literature pertaining to tourism's capacity as an instrument of economic development. I addition, progress could be made in the analysis of other factors that influence the relationship between tourism and economic development, such as the provision of infrastructure or security at the destination.

**Author Contributions:** Conceptualization, P.J.C.-G. and A.A.-O.; methodology, P.J.C.-G.; validation, P.J.C.-G.; formal analysis, A.A.-O.; investigation, P.J.C.-G. and A.A.-O.; resources, A.A.-O.; writing—original draft preparation, A.A.-O.; writing—review and editing, P.J.C.-G.; visualization, P.J.C.-G.; supervision, P.J.C.-G.; project administration, P.J.C.-G.; funding acquisition, A.A.-O. All authors have read and agreed to the published version of the manuscript.

**Funding:** This research received no external funding.

**Institutional Review Board Statement:** Not applicable.

**Informed Consent Statement:** Not applicable.

**Data Availability Statement:** No new data were created or analyzed in this study. Data sharing is not applicable to this article.

**Conflicts of Interest:** The authors declare no conflict of interest.

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
