# Peer review of "Tourism and Development: The Impact of Sustainability—Comparative Case Analysis"

_sustainability, doi:10.3390/su15021310_

Round 1

Reviewer 1 Report

Dear all,

Thanks for the invitation to review this work. In fact, tourism can considerably contribute to our economies and regional development as a critical factor for its success. Therefore, even if this study is brief, it could add and emphasize once more the relevance of this activity for the regional economies and consequent development.

In this regard, I recommend the acceptance of this manuscript.

Best Regards,

Author Response

We appreciate the reviewer's kind comments. We agree with the reviewer that it is important to analyze tourism as a tool with capacity for development.

Reviewer 2 Report

The quality of paper is not appropriate for this journal. It lacks scientific importance from many different points of view. The study is not correctly designed and the analysis lacks the highest scientific standards. Methods, tools software are not described with any details in order to allow anyone to draw conclusions and reproduce results. There is a complete gap in methodological part of the paper, so all other weakness of the manuscript (and there are many of them) seems less important ant comparing to this segment. In my opinion, the paper should be rejected because it is not possible to improve it in a way that is deserves publishing in this journal. 

Author Response

We regret the consideration of the reviewer regarding the article that we have presented. It must be taken into account that this is a review paper and it is not a research paper, so the methods and tools are more limited than in a research paper and, on the other hand, no type of of software. Finally, we hope that according to the changes we have introduced based on the comments of the other reviewers, you may change your opinion regarding the evaluation of the paper.

Reviewer 3 Report

The paper is interesting and actual.

However, regarding "structure" it should have a "conclusion". The conclusion is included in the section "Discussion" and it could be improved.

Some minor spell check required (for indstance: lines 81, 107, 265, ...)

A) Abstract and Introduction:

The objectives are enunciated. Authors point out: "the objective is to identify (...) if factors related to the environmental dimension of sustainability can be considered determining factors in the transformation of tourism growth (...) into economic development". However in the last two paragraphs of the paper (conclusion) the results should be better clarified in order to answer the paper questions.

Simultaneously, in the abstract authors state (lines 16 and 17): "if sustainability influences the expansion of tourism". Please clarify: are we discussing the sustainable development of tourism? And if a sustainable development (based on the 3 pillars: social, economic and environmental) could enhance tourism growth and development?

B) Theoretical framework 

Authors point out the importance of tourism related to 2019. (line 31) . They also should consider more recent data (see WTTC, 2022, Tourism Economic Impact - world). Authors should point out the huge impacts of Covid 19, that contribute to major challenges in tourism planning and management.

The paper points out relevant bibliography (for instance: Brundtland Report 1987  _ line 113; UN (2007)). However, it could be important to point out documents such as: Agenda 2030 (UN, 2015), Journey 2030 (UNWTO, 2017), Baseline Report on the Integration of Sustainable Consumption and Production Patterns into Tourism Policies (UNWTO, 2019). The last document, for "environmental impact" states the importance of impact areas such as "CHG emissions reduction", "reduction of air, soil and water pollutants", "Biodiversity conservation and sustainable land uses". These areas are linked with SDGs.

3. Materials and Methods

Authors say (line 140) that they considered "the last ten years of research": However it's not clear how and why authors selected 6 papers (6 case studies) (presented in tables 4.1 and 4.2). Please clarify.

Authors should also clarify, why they have selected case studies so different, in terms of territorial scale, years, dimension, etc. SEe:  papers selected discuss: a set of countries (144 countries) (line 157), EU-28, China, Colombia, United States (line 200), 139 countries (line 200). These differences should be better discussed.

4. Results

Regarding Case Study 3, (line 157) is about China (destination). However, in lines 186 and 187 authors discuss EU-28. 

Are Case studies a brief paper description? Should some paramenters of analysis be considered to enhance better conclusion?

5. Discussion

Sometimes authors speak about "determinants", but also about "factors", "environmental variables". Please clarify

Author Response

Dear reviewer:
We have proceeded to include and respond to all the changes proposed in the paper, which has been uploaded again to the application. With the proposed changes, the quality of the articles is now higher, thanks to the relevance, rigor and excellence of the reviewers' comments.
Details of the changes made based on your feedback are included in an attached file.
We await the final verdict, which we hope will be satisfactory since all the changes that had been proposed have been included.

Reviewer 4 Report

Review Report

3. Materials and Methods

R: This section has no validity in the study. The authors have to describe in detail the process of searching and collecting in the databases. What criteria were used in the search? How was the analysis process developed? What is the theoretical framework followed? They must follow a bibliometric approach, it is suggested that the authors research articles that use this type of methodology in their approaches (for example one that may help ground your study's methodology is a recent article with bibliometric methodology: "Virtual Reality in Tourism Promotion: A Research Agenda Based on A Bibliometric Approach").

5. Discussion

R: The authors are not doing a real discussion. At this point it is expected that the authors try to confront their results with previous studies, that is, if these results are in line with what is already known from other studies. It is necessary to do this reflection exercise. The study also needs a conclusion, theoretical and practical implications, study limitations and future recommendations.

Author Response

(The authors gave the same response as above.)

Round 2

Reviewer 2 Report

The manuscript is improved according to the instructions. 

Regards, 

Anđela

Author Response

Indeed, with the comments of all the reviewers in the previous stage, the article has improved considerably.

Thank you very much for the comments received.

Reviewer 4 Report

The authors are to be congratulated for a significant improvement of the paper. The paper has already been corrected and the gaps have been corrected.

Author Response

(The authors gave the same response as above.)
